# Adaptive Sampling for Discovery

**Ziping Xu**
Department of Statistics
University of Michigan
zipingxu@umich.edu

**Eunjae Shim**
Department of Chemistry
University of Michigan
eunjae@umich.edu

**Ambuj Tewari**
Department of Statistics
University of Michigan
tewaria@umich.edu

**Paul Zimmerman**
Department of Chemistry
University of Michigan
paulzim@umich.edu

## Abstract

In this paper, we study a sequential decision-making problem, called Adaptive Sampling for Discovery (ASD). Starting with a large unlabeled dataset, algorithms for ASD adaptively label the points with the goal to maximize the sum of responses. This problem has wide applications to real-world discovery problems, for example drug discovery with the help of machine learning models. ASD algorithms face the well-known exploration-exploitation dilemma. The algorithm needs to choose points that yield information to improve model estimates but it also needs to exploit the model. We rigorously formulate the problem and propose a general information-directed sampling (IDS) algorithm. We provide theoretical guarantees for the performance of IDS in linear, graph and low-rank models. The benefits of IDS are shown in both simulation experiments and real-data experiments for discovering chemical reaction conditions.

## 1 Introduction

Machine Learning (ML) models are becoming increasingly popular for discovery problems in various areas like drug discovery [40, 5, 38] and scientific discoveries [13, 23, 19]. Despite its success in real-data applications, the theoretical aspects of the problem are not fully understood in the machine learning literature. In this paper, we consider a particular case of discovery and formalize it as a problem that we call Adaptive Sampling for Discovery (ASD). In ASD, an algorithm sequentially picks $T$ samples $X_1, \ldots, X_T$ out of a size-$n$ unlabeled dataset $S^n$ ($T \ll n$) and obtains their labels $Y_1, \ldots, Y_T$ with the goal to maximize the sum $\sum_{t=1}^{T} Y_t$. The labels correspond to the quality of discoveries, which can be continuous or ordered categorical variables depending on the types of discoveries. As a sequential decision-making problem, we find it substantially connected to the existing literature on Multi-arm Bandit (MAB) and other related decision-making problems, which we briefly review in the rest of this section.

**Multi-arm bandit.** MAB is a decision-making problem that maximizes the total rewards by sequentially pulling arms and each arm corresponds to a reward distribution [27]. MAB faces the well-known exploration–exploitation trade-off dilemma. ASD has to deal with the same exploration-exploitation dilemma if one treats each arm as an unlabeled point. The major difference between ASD and MAB is that each arm after being pulled once can still be pulled for infinite number of times, while we can only label a sample once in ASD. This is because a discovery can only be made once: one receives no credit making the same discovery twice.

36th Conference on Neural Information Processing Systems (NeurIPS 2022).

In fact, as we will show later, ASD can be an MAB when each unique value in $S^n$ has larger than $T$ repeated samples and $n \ll T^2$. Moreover, a variant of MAB called Sleeping Expert [22, 24] can be seen as a more general problem setup of ASD. One can intuitively interpret Sleeping Expert as the bandit problem where the set of available arms varies. More generally, bandit algorithms that allows adversarial choice of the action set can be used to solve the ASD problem. However, due to its generality, the regret bounds developed in [24] are loose or vacuous in our setup. Detailed discussions are deferred to Section 2.

Due to the similarities between bandits and ASD, we adopt the same performance measurement, regret, from bandit literature. Regret is the highest sum of labels that could have been achieved minus the actual sum of labels.

There is a long line of work in the bandit literature on solving the exploration-exploitation tradeoff. Popular methods includes UCB (upper confidence bound), TS (Thompson sampling). UCB keeps an optimistic view on the uncertain model [4, 26], which encourages the algorithm to select arms with higher uncertainty. TS [1] adapts a Bayesian framework, which samples a reward vector from its posterior distribution and select arms accordingly. In this paper, we adopt another family of methods called Information-directed Sampling (IDS). IDS [33] balances between the expected information gain on the unknown model and instant regret with respect to the posterior distribution under a Bayesian framework. IDS has been shown to outperform UCB in problem with rich structural information, e.g. sparse linear bandit [14]. As we show below (Proposition 1), the ASD problem is not interesting in the unstructured case which means we have to use structural assumptions. That makes IDS a natural choice.

Traditional MAB algorithms designed for finite-armed bandits are not applicable to ASD since the number of unlabeled samples $n$ can be much larger than horizon $T$. There are works considering infinite-armed bandits by assuming structural information across arms. For instance, linear bandit assumes a linear model for the reward generation [27], [41] assumes Lipschitz-continuity and [42] considers neural network model for reward generation. However, there are few works studying sleeping expert with structure across arms beyond linear structure.

**Other related literature.** With a similar adaptive sampling procedure, Active Learning (AL) aims at achieving a better *model accuracy* with a limited number of labels. As the reader may have noticed, the goal of AL aligns with ASD in the early stage when the model has high uncertainty and the predictions are high unreliable, while in the later stage ASD aims at better discovery performance. Indeed, there are active learning algorithms based on the idea of maximum information gain [3]. [29] applies the idea to the matrix completion problem, which significantly improves the prediction accuracy from random selection. Other works with a similar goal go by the name *sequential exploration* [6, 7, 17]. Their works lack a theoretical justification and the algorithms are not generally applicable.

A more relevant setup in the active learning literature is Active Search (AS). Active search is an active learning setting with the goal of identifying as many members of a given class as possible under a labeling budget. AS concerns the Bayesian optimal methods named Efficient Nonmyopic Search (ENS) [21, 20, 32], which is not computationally efficient. Their approximate algorithms for the Bayesian optimal method do not provide strong theoretical guarantee. A more detailed comparison between our proposed approach and ENS is given in Appendix I.

**Main contributions.** In this paper, we formulate the adaptive sampling for discovery problem and propose an generic algorithm using information-direct sampling (IDS) strategy. We theoretically analyze the regret of IDS under generalized linear model, low-rank matrix model and graph model assumptions. Indeed, the analysis for linear model are directly from IDS for linear bandit. The regret analysis for low-rank model and graph model are new even in the bandit literature. The results are complemented by simulation studies. We apply the algorithms to the real-world problems in reaction condition discovery, where our algorithm can discover about 20% more plausible reaction conditions than random policies and other simple baseline policies adapted from the bandit literature.

## 2   Formulation

In this section, we formally formulate ASD and rigorously discuss its connections to MAB.

**Notations.** We first introduce some notations that are repeatedly used in this paper. For any finite set $S$, we let $\mathcal{D}(S)$ be the set of all the distributions over the set. For any positive integer $n$, we let $[n] = \{1, \ldots, n\}$. We use the standard $O(\cdot)$ and $\Omega(\cdot)$ notation to hide universal constant factors. We also use $a \lesssim b$ and $a \gtrsim b$ to indicate $a = O(b)$ and $a = \Omega(b)$. ASD is a sequential decision-making problem over discrete time steps. In general, we let $\mathcal{F}_t$ be the observations up to decision step $t$. We adapt a Bayesian framework and let $\mathbb{E}_t[\cdot] = \mathbb{E}[\cdot \mid \mathcal{F}_t]$ be the conditional expectation given the history up to $t$. Let $\mathbb{P}_t$ be the probability measure of posterior distribution.

**Adaptive sampling for discovery.** We formulate ASD problem as follows. Consider a problem with the covariate space $\mathcal{X} \subset \mathbb{R}^d$ and label space $\mathcal{Y} \subset [0, 1]$. For categorical discoveries, the labels are ordered such that higher labels reflects better discoveries. Given a input $x \in \mathcal{X}$, its label is generated from a distribution $\mathcal{D}_{\theta|x}$ over $\mathcal{Y}$ parametrized by unknown parameters $\theta$. We adopt a Bayesian framework and denote the prior distribution of $\theta$ by $\phi$. Given a sequence of $t-1$ observations $\mathcal{F}_t = \{(X_i, Y_i)\}_{i=1}^{t-1}$, the posterior distribution of $\phi$ is denoted by $\phi(\cdot \mid \mathcal{F}_t)$. We also let $f_\theta(x) = \mathbb{E}_{Y \sim \mathcal{D}_{\theta|x}}[Y]$ be the expected label of input $x$ under model $\theta$.

ASD starts with an unlabeled dataset $S^n \subset \mathcal{X}$ with $n$ samples. Our goal is to adaptively label $T$ samples $\{X_1, \ldots, X_T\} \subset S^n$ to maximize the sum of labels $\sum_{t=1}^T Y_t$. The discovered set is removed from available unlabeled set, i.e. each arm can only be pulled once. At each step $t$, an algorithm chooses an input $X_t$ from $S_t^n := S_n \setminus \{X_1, \ldots, X_{t-1}\}$. The environment returns the label $Y_t \sim \mathcal{D}_{\theta|X_t}$.

To measure the performance of any algorithm, we borrow the definition of Bayesian regret from bandit literature [27]. Note that the regret definition is also used in Sleeping Expert [22], which is a more general framework than ASD. With respect to any chosen random sequence $(X_1, \ldots, X_T)$, we define regret by

$$R_T = \max_{x_1, \ldots, x_T \in S^n} \sum_{t=1}^T f_\theta(x_t) - \sum_{t=1}^T f_\theta(X_t). \tag{1}$$

Let $S_t^n = S^n \setminus \{X_1, \ldots, X_{t-1}\}$ be the remaining unlabeled set at the start of step $t$. An algorithm $\mathcal{A}$ is a sequence of policies $(\pi_1, \ldots, \pi_T)$. Let $\mathcal{D}(S_t^n) \subset \mathbb{R}^{n-t-1}$ be the set of all the distributions over $S_t^n$ that are denoted as $n-t-1$ vectors. Each $\pi_t$ is a mapping from history $\mathcal{F}_t$ to a distribution $\mathcal{D}(S_t^n)$ over $S_T^n$. The Bayesian Regret w.r.t. an algorithm is

$$\mathcal{BR}(T, \mathcal{A}) := \mathbb{E}R_T,$$

where the expectation is taken over the randomness in the choice of input $X_t$, the labeling $Y_t$ and over the prior distribution on $\theta$. Let $X_1^*, \ldots, X_T^*$ be the sequence that realizes the maximum in (1). Note that $X_1^*, \ldots, X_T^*$ are random variables depending on $\theta$. In general, the goal the algorithm is to achieve a sublinear regret upper bound.

**Connections to bandit problem.** This problem degenerates to a classical bandit problem when $\mathcal{X}$ is discrete and $S^n$ has infinite number of unlabeled samples for each unique value in $\mathcal{X}$. In the bandit literature some frequentist results are available. The problem has the minimax regret rate $\sqrt{|\mathcal{X}|T}$. However, in our problem, we generally consider large unlabeled set ($n \gg T$). In fact, when $|\mathcal{X}|$ is larger than $T^2$, Proposition 1 shows that a linear regret is unavoidable without a structural assumption. Therefore, we discuss different types of structural information that allows information sharing across different inputs.

**Proposition 1** (Lower bound for nonstructural case). *Consider the following set of ASD problems. Let $n = T^2$, $S^n = [n]$ and $\mathcal{D}_{\theta|x} = \mathcal{N}(\theta_x, 1)$ for $x \in [n]$ and $\theta \in [0, 1]^n$. For any algorithm $\mathcal{A}$, there exists some prior distribution over the set of ASDs described above such that $\mathcal{BR}(T, \mathcal{A}) \gtrsim T$.*

**Compared with sleeping expert.** Sleeping expert considers a generalized bandit framework where the available (awake) action set $\mathcal{A}_t$ are changing. It is the same as ASD except for that the changes of the unlabeled dataset (or awake action set) follow certain dynamic. That is whenever an input is labeled, it is taken out of the unlabeled dataset. However, sleep expert normally assumes a stochastic distribution on $\mathcal{A}_t$ or adversarial changes [24, 9], which makes their regret bound loose. To be specific, [22] derives regret bounds for unstructured bandits, which leads to vacuous regret bound of $\sqrt{nT}$. For structured case, [24] derives the same regret bound in this paper, due to the strong

structure assumption for linear model. There are few literature for sleeping bandits with complicated structures beyond linear model. The regret bounds in the bandit literature for the other two models, graph and low-rank matrix models, are not applicable both resulting a $\Omega(T)$ regret bound, which assures the necessity of studying ASD problem by itself.

# 3  Information-directed Sampling and Generic Regret Bound

In this section, we introduce a strategy called Information-Directed Sampling and develop a generic regret bound. IDS runs by evaluating the information gain on the indices of the top $T$ inputs with respect to the labeling of a certain input $x$. It balances between obtaining low expected regret in the current step and acquiring high information about the inputs that are worth labeling. We first introduce *entropy* and *information gain*.

**Definition 1** (Entropy and information gain). *For any random variable $X \in \mathcal{X}$ with a density function $\mathbb{P}$, its entropy is defined as $H(X) = -\int_{x \in \mathcal{X}} P(x) \log(P(x))$. Let $H(X \mid Z)$ be the conditional entropy of $X$ given $Z$. The mutual information between any two random variables $Y$ and $X$ can be defined as $I(X; Y) = H(X) - H(X \mid Y)$. Conditional mutual information is $I(X; Y \mid Z) = H(X \mid Z) - H(X \mid Y, Z)$. Furthermore, we use $H_t$ and $I_t$ for the entropy and mutual information under the posterior distribution up to step $t$.*

For IDS, we let $\Delta_t(x) = \mathbb{E}_t[\max_{x' \in S_t^n} f_\theta(x') - f_\theta(x)]$ be the expected instant regret. Let the information gain of selecting $x$ at the step $t$ be $g_t(x) = I_t(X_{t,1}^*, \ldots, X_{t,T-t+1}^*; Y_t \mid \mathcal{F}_{t-1}, X_t = x)$, where $(X_{t,1}^*, \ldots, X_{t,T-t+1}^*)$

$$
= \left\{ (x_1, \ldots, x_{T-t+1}) : x_i \in S_t^n, f_\theta(x_1) \geq \cdots \geq f_\theta(x_{T-t+1}) \geq \max_{x' \neq x_1, \ldots, x_{T-t+1}} f_\theta(x') \right\},
$$

are the random variable for the top $T - t + 1$ unlabeled points. Note that $\{X_{t,1}^*, \ldots, X_{t,T-t+1}^*\} \subset \{X_1^*, \ldots, X_T^*\}$. Intuitively, $g_t(x)$ captures the amount of information on the indices of the top $T - t + 1$ points gained by labeling $x$.

IDS minimizes the following ratio

$$
\pi_t^{\text{IDS}} \in \arg\min_{\pi \in \mathcal{D}(S_n^t)} \Psi_{t,\lambda}(\pi) := \frac{(\Delta_t^T \pi)^\lambda}{g_t^T \pi}, \text{ for some constant } \lambda > 0,
$$

where $\Delta_t \in \mathbb{R}^{n-t+1}$ and $g_t \in \mathbb{R}^{n-t+1}$ are the corresponding vectors for the expected single-step regret and information gain. Constant ratio here weighs between instant regret and information gain. A higher $\lambda$ prefers points with lower instant regrets. An abstract algorithm is given Algorithm 1.

---

**Algorithm 1** IDS (Information-Directed Sampling) for Discovery

---

**Input:** Unlabeled dataset $S^n$, prior distribution $\phi$, total number of steps $T$, constant $\lambda$.
Initialize history $\mathcal{F}_0 = \{\}$, $S_1^n = S^n$.
**for** $t = 1$ **to** $T$ **do**
    Calculate $\Delta_t(x)$ and $g_t(x)$ for each $x \in S_t^n$ using posterior $\phi(\cdot \mid \mathcal{F}_{t-1})$.
    Sample $X_t \sim \arg\max_\pi \Psi_{t,\lambda}(\pi)$ and receive label $Y_t$ from $\mathcal{D}_{\theta \mid X_t}$.
    Update history $\mathcal{F}_t = \mathcal{F}_{t-1} \cup \{(X_t, Y_t)\}$ and unlabeled dataset $S_{t+1}^n = S_t^n \setminus \{X_t\}$.
**end for**

---

We can show that the following generic Bayesian regret upper bound holds.

**Lemma 1.** *Suppose $\pi^{\text{IDS}} = (\pi_t)^{t \in [T]}$ and $\Psi_{t,\lambda}(\pi_t) \leq \Psi_{*,\lambda}$. Then we have IDS using constant $\lambda$ has the following regret bound*

$$
\mathcal{BR}(T, \text{IDS}) \lesssim (\Psi_{*,\lambda} H(X_1^*, \ldots, X_T^*) T^{\lambda-1})^{1/\lambda}.
$$

As the case in the bandit problem, the regret bound depends on the worst-case information ratio bound and the entropy of the random variables for the top $T$ unlabeled points. Now we will discuss how to bound information ratios under various structural information assumptions.

## 4 Bounding Information Ratio under Structural Assumptions

In this section, we study three cases with different structural information.

### 4.1 Generalized linear model.

In this subsection, we discuss a generalized regression model, which includes logistic regression, corresponding to a natural binary discovery problem. Consider $d$-dimensional generalized linear regression problem. The label $Y$ given an input $X$ is generated by

$$Y = \mu(X^\top \theta^*) + \epsilon,$$

where $\mu$ is usually referred as link function. We assume that $Y$ is uniformly bounded in $[0, 1]$. In addition, we assume an upper bound on the derivatives of $\mu$.

**Assumption 1.** *The first order derivatives of $\mu$ are upper-bounded by $L_\mu$.*

**Theorem 1.** *Under Assumption 1, we have $\Psi_{*,2} \leq L_\mu d/2$, which gives the Bayesian regret bound*

$$\mathcal{BR}(T, IDS) \lesssim \sqrt{dH(X_1^*, \ldots, X_T^*)L_\mu T} \tag{2}$$

A naive bound for $H(X_1^*, \ldots, X_T^*)$ in [33] does not apply here because it introduces a $\log(n)$ dependency. Instead, we notice that $X_1^*, \ldots, X_T^*$ is a deterministic function of $\theta$ and $S^n$. We can bound the entropy of $\theta$ instead. Assuming a Gaussian prior on $\theta$, $H(\theta) \lesssim d \log(d)$.

Indeed, the regret in (2) is analogous to the regret bound in linear bandit [33, 27]. Theoretically, IDS does not show an advantage. This is partly because carefully designed algorithms for linear model are sufficiently utilizing the structural information efficiently. [14] has shown that IDS achieves a much tighter lower bound for sparse linear bandit compared to UCB-type algorithms and Thompson sampling algorithms. We show in the Appendix G that IDS enjoys a regret bound of $\tilde{\mathcal{O}}(sT^{2/3})$ in the discovery setting with a sparse linear structure, with $s$ being the sparsity.

### 4.2 Low-rank matrix

In this subsection, we discuss a novel problem setup: low-rank matrix discovery. Low-rank matrix completion problem has been extensively studied in the literature [25, 18, 31]. However, most of the algorithms select entries with a uniform random distribution. In many applications, appropriately identifying a subset of missing entries in an incomplete matrix is of paramount practical importance. For example, we can encourage users to rate certain movies for movie rating matrix recovery. While selecting the users that reveal more information is important, it is also important to push the movies aligning with users tastes. Active matrix completion has long been studied in the literature [8, 29, 15]. As far as we know, no work has considered a discovery setup, where the goal of adaptive sampling is not only to get a better recovery accuracy but also to maximize the sum of observed entries.

**Formulation.** Consider an unknown matrix $Y \in \mathbb{R}^{m_1 \times m_2}$, whose entries $(i, j) \in [m_1] \times [m_2]$ are sampled in the following manner:

$$Y_{i,j} = e_i^T M_{i,j} e_j + \epsilon_{i,j} \text{ for } \epsilon_{i,j} \text{ being the noise for entry } i, j,$$

where $M \in \mathbb{R}^{m_1 \times m_2}$ is the unknown rank-$r$ matrix and $r \ll \min\{m_1, m_2\}$. The matrix $M$ admits SVD, i.e. $M = UDV^T$, where $U \in \mathbb{R}^{m_1 \times r}$ and $V \in \mathbb{R}^{r \times m_2}$ are orthogonal matrix and $D \in \mathbb{R}^{r \times r}$. For simplicity, we let $m_1 = m_2 = m$. Our analysis can be easily extended to the case $m_1 \neq m_2$.

Note that this setup is closely connected to the low-rank bandit problem except for that the actions are standard basis and can only be selected once. The best results in the literature [28] achieve a regret bound of $\tilde{\mathcal{O}}(m^{3/2}\sqrt{rT})$. Since in our case, $T \leq m^2$, the regret bound becomes vacuous. This finding implies that we need some further assumptions on the structure of the matrix. A common assumption in the matrix completion is *incoherence*, an important concept that measures how the subspace aligns with the standard basis.

**Definition 2** (Coherence). *The coherence of subspace $U$ with respect to the standard $i$-th basis vector $e_i$ is defined as $\mu_i(U) = \|P_U e_i\|_2^2$, where $P_U$ is the orthogonal projection onto the subset defined by $U$. Note that $\mu_i(U) = \|U^T e_i\|_2^2$ as well.*

**Assumption 2** (Incoherence)**.** *There exists some constant $\gamma > 0$, such that the unknown matrix $M$ is incoherent, i.e. $\max_{i \in [m]} \mu_i(U) \leq \sqrt{\gamma r/m}$ and $\max_{i \in m} \mu_i(V) \leq \sqrt{\gamma r/m}$.*

To introduce our results, we further define some constants.

**Remark 1.** *Let $\bar{d}$ be the maximum singular value of the unknown matrix $M$. We have*

$$\max_{i,j} |M_{i,j}| \leq \max_{i,j} \mu_i(U)\mu_j(V)\|D\|_F \leq \gamma r^{3/2}\bar{d}/m \coloneqq B.$$

**Theorem 2.** *Under the low-rank matrix model with Assumption 2, the worst-case information ratio can be bounded by $\Psi_{*,3} \leq 4B(B^2 + 1)r^3\gamma^2$, which gives us a regret bound of*

$$\mathcal{BR}(T, \text{IDS}) \lesssim \left(4B\left(B^2 + 1\right)r^3\gamma^2 H(M)T^2\right)^{1/3}.$$

The entropy term $H(M)$ depends on the distribution of $M$. To give an example, let $M = UV^T$, where $U, V^\top \in \mathbb{R}^{m \times r}$ and each element in $U$ and $V$ are sampled independently from standard Gaussian. Then $H(M) \leq H((U, V)) = H(U) + H(V) \lesssim mr\log(mr)$. In general, the regret bound is still sublinear when $mr^4 \ll T \ll m^2$.

## 4.3 Graph

We consider the discovery problem with graph feedback. Specifically, we are given a graph $G = (S^n, E)$, where the unlabeled set $S^n$ is the node set. By labeling a node $x \in S^n$, we receive noisy outcomes for all the nodes connecting to $x$. Let the side information at step $t$ be $O_t(X_t) = \{\tilde{Y}_{x'}\}_{x':(x',X_t) \in E}$, where $\tilde{Y}_{x'} = x' + \epsilon_{x'}$ for $\epsilon_{x'}$ being the zero-mean noise given $x$ being selected. In addition to the side information, we also receive the label for $X_t$, which is sampled in the same manner. This setup is also studied in the sleeping expert literature [9]. [9] gives a regret bound depending on $\mathbb{E}[\sum_x T_x/Q_x]$, where $T_x$ is the total number of visits on action $x$ and $Q_x$ is the total number of observations for action $x$. The ratio can be low when the algorithms tend to select nodes with high degrees and their algorithms are not designed for that purpose. Therefore, the structure are not sufficiently exploited and the regret bound can be loose.

To measure the complexity of a graph, we introduce the following two definitions.

**Definition 3** (Maximal independent set)**.** *An independent set is a set of vertices in a graph such that no two of which are adjacent. A maximal independent set (MIS) is an independent set that is not a subset of any other independent set. We denote the cardinality of the smallest maximum independent set of a graph $G$ by $\mathcal{C}(G)$.*

**Definition 4** (Clique cover number)**.** *A Clique of a graph $G = (\mathcal{K}, \mathcal{E})$ is a subset $S \subseteq \mathcal{K}$ such that the sub-graph formed by $S$ and $\mathcal{E}$ is a complete graph. A Clique cover of a graph $G = (\mathcal{K}, \mathcal{E})$ is a partition of $\mathcal{K}$, denoted by $\mathcal{Q}$, such that $Q$ is a clique for each $Q \in \mathcal{Q}$. The cardinality of the smallest clique cover is called the clique cover number, which is denoted by $\chi(G)$.*

**Remark 2.** *Clique cover number is guaranteed to be larger than the cardinality of the smallest maximal independent set, i.e. $\mathcal{C}(G) \leq \mathcal{X}(G)$.*

Since the nodes in the graph are diminishing, we further define a stable version of MIS.

**Definition 5.** *Let $\mathcal{G}_t(G)$ be the set of all subgraphs of $G$ with $N - t$ nodes and define $\mathcal{C}_t(G) = \max_{G_t \in \mathcal{G}_t(G)} \mathcal{C}(G_t)$.*

**Assumption 3.** *We assume that $x + \epsilon_x \leq B$ almost surely for all $x \in S^n$ and some constant $B > 0$.*

**Theorem 3.** *Under Assumption 3, the Bayesian regret of the IDS with $\lambda = 2$ can be upper bounded by*

$$\mathcal{BR}(T, IDS) \lesssim \min\{(B\mathcal{C}_T(G)T)^{2/3}, (\mathcal{X}(G)T)^{1/2}\}.$$

Let us discuss certain graphs where the regret bound can be low. First, if the graph can be decomposed into $K \ll T$ complete subgraphs, then $\mathcal{X}(G) \leq K$ and we have a regret bound of $\sqrt{KT}$. An example, in which $\mathcal{C}(G)$ is low while $\mathcal{X}(G)$ is high, is a star-shaped graph, with a set of nodes in the center of the graph connecting to all the other nodes. See Figure 2 in the Appendix for an illustration.

### 4.4 A generic results for models with structural information

As shown in sections 4.2 and 4.3, a $T^{2/3}$ regret upper bound with a small coefficient can be derived when the model has certain structural information. We give the following generic result, stating that as long as there exists a policy $\mu$, whose information gain can be lower bounded by the instant regret of Thompson sampling policy an upper bound for $\Psi_{\star,3}$ can be derived.

**Proposition 2.** *Assume that there exists a policy $\mu$ such that $g_t^\top \mu \geq \phi(\Delta_t^\top \pi_t^{ts})^2$, where $\pi_t^{ts}$ is the Thompson sampling policy at the step $t$ for some constant $\phi$ and assume that the instant regrets are uniformly bounded by $B$. Then $\Psi_{\star,3} \leq 2B/\phi$, which gives a regret bound of $\mathcal{O}((BTH(\theta)/\phi)^{2/3})$.*

## 5 Experiments

We start with simulation studies on the three problems: (generalized) linear model, low-rank matrix and graph model. Since information gain can be hard to calculate in some problems, we introduce sampling-based approximate algorithms, which generate random samples from the posterior distribution and evaluate a variance term instead of the original information gain. We will also introduce general Thompson Sampling algorithms for all the three problem setups.

### 5.1 Approximate algorithms

It may not be efficient to evaluate the information gain in certain problem setups. For example, one way to generate random low-rank matrix is to generate its row and column spaces from Gaussian distribution. We do not have a closed form for the posterior distribution of the resulting matrix. Neither can we evaluate the information gain. To that end, we follow an approximate algorithmic design in [33], which replaces the information gain with a conditional variance $v_t(x) = \text{Var}_t(\mathbb{E}[Y_{t,x} \mid X_1^*, X_t = x])$, which will be evaluated under random samples from posterior distribution using MCMC. The conditional variance is a lower bound of the information gain. An approximate algorithm is given by Algorithm 2 in Appendix H.

**Proposition 3.** *The following lower bound for information gain holds, $g_t(x) \geq v_t(x)$.*

**Compared algorithms.** For the rest of section, we compare IDS (the approximate algorithm), TS, UCB (if the bandit version is available) and random policy. As a method that is closely connected to IDS, Thompson sampling (also referred as posterior sampling) is also compared in our experiments. Thompson sampling [36] at each round samples a $\theta_t$ from its posterior distribution and then selects the input with lowest instant regret w.r.t to $\theta_t$. TS does not account for the amount of information gain, thus can be insufficient in utilizing the structural information. UCB is an important algorithm design in bandit literature. Though it is not specifically designed for ASD, we apply UCB to (generalized) linear ASD by manually removing arms that have been selected.

### 5.2 Simulation studies

**Linear and logistic regression model.** The data generating distribution are specified below. Unlabeled dataset is sampled from $\mathcal{N}(0, \sigma_x^2)$ and the prior distribution of $\theta^*$ is $\mathcal{N}(0, \sigma_\theta^2)$. The noise $\epsilon$ is sampled from $\mathcal{N}(0, \sigma_\epsilon^2)$. We study the effects of $d = 20, 50, 100$. The same setup is used for logistic regression model. The posterior distribution for $\theta$ in simple linear regression is also Gaussian. For logistic regression, we used a Laplace approximation for the posterior distribution [16]. We use the GLM-UCB [12] for logistic regression.

**Graph model.** We consider different types of graphs. For each graph, the node values are sampled from standard Gaussian distribution and the noise are sampled from Gaussian with zero mean and $\sigma_\epsilon^2$ variance. We tested $\sigma_\epsilon^2 = 0.1, 1.0, 10$. We experiment on graphs with $N = 900$. For a better demonstration of the early stage performance, we only show the first 50 steps. We experiment the following types of graphs. 1) Random connections: any node has a probability of $p \ (= 0.01)$ to be connected; 2) Complete graph: every pair of nodes has an edge; 3) Star graph: $2/3$ of nodes are randomly connected with a probability of $p \ (= 0.01)$. And the rest of $1/3$ nodes are connected to all the nodes. Note that an example of star graph is given in Figure 2. Since, no node in complete graph provides more information than others, we expect TS and IDS perform the same. We expect IDS

outperforms TS more significantly in star graphs than it does in random graphs as there are more nodes gaining global information.

**Low-rank matrix** We sample each entry of $U \in \mathbb{R}^{d \times r}$ and $V \in \mathbb{R}^{d \times r}$ from $\mathcal{N}(0, \sigma_0^2)$ and let the unknown matrix $Y = UV^T + \epsilon$, with each entry of $\epsilon$ from $\mathcal{N}(0, \sigma_\epsilon^2)$. Since we do not have a closed form for the posterior distribution, posterior samples are generated from a MCMC algorithm. In the experiment, we let $m = 30$, which gives us 900 steps at maximum. For a better demonstration of the early stage, we show the cumulative regrets in the first 100 steps.

**Results.** Figure 1 shows the simulations results for the four models above. IDS has lower cumulative regrets over all the steps in linear model, logistic model, graph random and graph star model. TS and IDS always outperform random policy and UCB (if available), showing a benefit of developing algorithms for ASD. In the graph experiments, IDS and TS performs almost the same for complete graph because any input gains information of the full graph. IDS has slightly lower regrets than TS in random graph because of the weak structural information. It has significantly lower regrets than TS and random because of the stronger structural information. As shown in (b, d), the cumulative regrets at steo 100 grows linearly with $d$. A higher noise also leads to a higher regret as shown in (f, h, j, l). IDS performs better in different levels of $d$ for linear models and different levels of $\sigma$ for graph random, graph star, matrix.

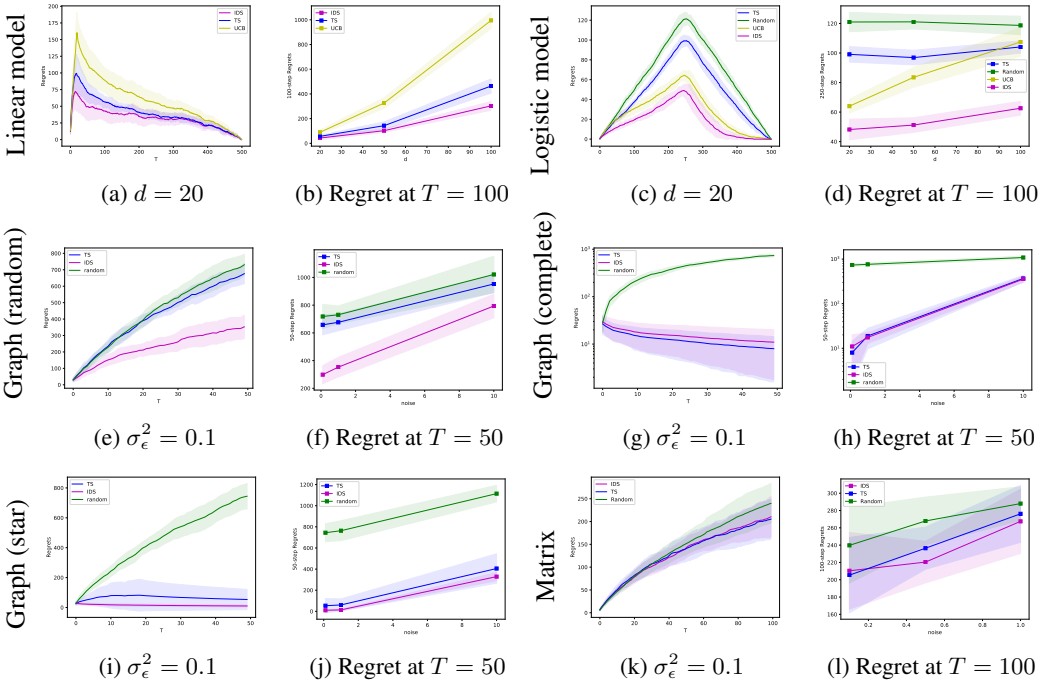

Figure 1: Cumulative regret curves for linear regression (a, b), logistic regression (c, d), random graph, complete graph, star graph (e-j) and low-rank matrix (k, l). Columns 1 and 3 are the cumulative regret curves for different models. Columns 2 and 4 are cumulative regret at an early step for different hyperparameters. For linear and logistic regression model, $d$ ranges in $\{20, 50, 100\}$. For graph model, $\sigma_\epsilon^2$ ranges in $\{0.1, 1.0, 10\}$. For low-rank matrix, $\sigma_\epsilon^2$ ranges in $\{0.1, 0.5, 1.0\}$.

## 5.3 Reaction condition discovery

To further test the usefulness of IDS and TS for ASD in real-world problems, we consider a *reaction condition discovery* problem in organic chemistry.

**Background and dataset.** In the chemical literature, a single reaction condition that gives the highest yield for one set of reacting molecules is often demonstrated on a couple dozen analogous

molecules. However, for molecules with structures that significantly differ from this baseline, one needs to *rediscover* working reaction conditions. This provides a significant challenge as the number of plausible reaction conditions can grow rapidly, due to factors such as the many combinations of individual reagents, varying concentrations of the reagents, and temperature. Therefore, strategies that can identify as many high-yielding reaction conditions as early as possible would greatly facilitate the preparation of functional molecules. Similar reaction condition discovery problem is studied under a transfer learning framework [30, 37].

The IDS algorithms may therefore be highly useful in examining experimental data compared to UCB, TS and random policies. Two experimental datasets on reactions that form carbon-nitrogen bonds, which are of high importance for synthesizing drug molecules, were therefore selected as specific test cases. The first dataset, named Photoredox Nickel Dual-Catalysis (PNDC) [11], comes from a reaction that involves two types of catalysts (a photocatalyst and a nickel catalyst) that work together to form carbon-nitrogen bonds. It has $n = 80$ reaction conditions for discovery. The variables in the dataset are the identity of photocatalysts, relative catalyst concentration, and relative substrate concentration, totaling 80 reaction conditions. The second C-N Cross-Coupling with Isoxazoles (CNCCI) dataset [2] comes from a collection of a cross-coupling reactions in the presence of an additive. This dataset was designed to test the reaction's tolerance to the additive (isoxazole) and includes 330 candidate reactions. Both datasets have continuous response variables (reaction yield).

**Results.** Table 1 demonstrated the results of linear IDS, TS, UCB for PNDC and CNCCI. We tuned hyperparameters for all three algorithms. Details are given in Appendix H. Over 17 steps, IDS showed a dominant performance in all 11 molecules tested in PNDC with an average of 1.58 more discoveries than random policy. Either IDS or TS performs the best in the 9 combinations of catalyst and base tested in CNCCI dataset. Over 66 steps, IDS discovers 14.1 more plausible reaction conditions on average than random.

Table 1: Summary of cumulative regrets at 20% of total number of steps. Each column corresponds to a target molecule. The regrets are averages of 10 independent runs, whose standard deviation are given in the brackets. Best algorithm for each column is marked in red. (-) denote that the algorithms are deterministic. The columns are the indices for target molecules, whose details are provided in Appendix H.

| Molecules\Algorithms | X2 | X3 | X4 | X5 | X6 | X8 |
|---|---|---|---|---|---|---|
| Random | 8.77 (1.44) | 7.74 (0.75) | 2.12 (0.20) | 1.70 (0.15) | 4.96 (0.72) | 9.90 (1.47) |
| IDS | 6.53 (0.20) | 5.70 (0.52) | 1.56 (0.06) | 0.81 (0.01) | 3.44 (0.81) | 9.35 (0.96) |
| TS | 7.64 (1.11) | 6.67 (1.09) | 2.15 (0.48) | 1.22 (0.38) | 4.07 (1.30) | 9.78 (1.69) |
| UCB | 9.63 (-) | 7.66 (-) | 2.58 (-) | 1.42 (-) | 5.26 (-) | 10.43 (-) |
| | X11 | X12 | X13 | X14 | X15 | |
| Random | 7.15 (1.21) | 5.27 (0.92) | 4.44 (0.43) | 5.72 (0.35) | 2.65 (0.47) | |
| IDS | 2.40 (0.12) | 3.95 (0.19) | 3.22 (0.21) | 4.94 (0.07) | 1.09 (0.03) | |
| TS | 2.90 (1.76) | 4.42 (0.90) | 4.79 (1.17) | 5.39 (0.64) | 1.35 (0.43) | |
| UCB | 3.58 (-) | 5.28 (-) | 4.53 (-) | 5.86 (-) | 2.03 (-) | |

(a) Results for PNDC

| Catalyst and Base \ Algorithms | L1+B1 | L2+B1 | L3+B1 | L4+B1 | L1+B2 |
|---|---|---|---|---|---|
| Random | 25.65 (2.11) | 23.08 (1.98) | 19.90 (1.19) | 15.96 (0.83) | 29.59 (1.26) |
| IDS | 15.55 (1.52) | 7.90 (0.28) | 10.03 (0.88) | 10.81 (0.79) | 10.76 (0.51) |
| TS | 11.84 (1.04) | 9.82 (2.23) | 11.58 (4.31) | 10.39 (0.81) | 12.49 (1.10) |
| UCB | 13.63 (-) | 10.08 (-) | 12.00 (-) | 11.98 (-) | 17.41 (-) |
| | L4+B2 | L1+B3 | L3+B3 | L4+B3 | |
| Random | 28.78 (2.59) | 19.84 (0.81) | 29.03 (2.22) | 27.08 (2.51) | |
| IDS | 10.51 (1.32) | 8.63 (0.97) | 8.76 (0.32) | 9.44 (0.52) | |
| TS | 12.68 (2.33) | 10.61 (0.99) | 12.44 (7.39) | 9.80 (2.39) | |
| UCB | 15.60 (-) | 12.42 (-) | 11.49 (-) | 12.32 (-) | |

(b) Results for CNCCI

# 6 Discussion

In this paper, we posed and comprehensively studied the Adaptive Sampling for Discovery problem and proposed the Information-directed sampling algorithm along with its regret analysis. The results are complemented with both simulation and real-data experiments. There are a few open directions following the results in the paper. First, although our paper shows bandit-like regret bounds, ASD problem by nature is different from bandit in the way that the unlabeled set is diminishing, which may further reduce the complexity of the problem. For example, in the linear model, all the unlabeled points may fall on a lower dimensional hyperplane, which would reduce the dependence on $d$. Second, frequentist regret may be studied instead of adopting a Bayesian framework. More practical algorithms e.g. random forests and neural network may be considered in the future work. Sampling from posterior can be slow for certain models. A fast and generalizable algorithm using IDS for more complicated and larger scale real-data applications may be considered.

# 7 Acknowledgement

We acknowledge the support of National Science Foundation via grant IIS-2007055 and CHE-1551994.

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
