# OpenReview forum: "Adaptive Sampling for Discovery"
_NeurIPS.cc/2022/Conference — NeurIPS 2022 Accept_

### Official Review · Reviewer_uuVE · 2022-07-09

**Rating:** 6
**Confidence:** 3
**Soundness:** 3 good
**Presentation:** 2 fair
**Contribution:** 3 good

**Summary:**

This paper presents an adaptive sampling strategy for labeling in the finite sample region (without replacement in contrast to MultiArmed Bandit (MAB) scenario) with the overall objective to maximize the sum of label values of sampled points. The proposed algorithm (IDS) uses  information gain (objective being ratio of regret and information gain at that time step) in order to sample the next point to label. The authors show leveraging model structure leads to improved regret bounds for low-rank matrix and feedback graph models. Finally, the authors present real-world application of IDS to the problem of reaction discovery.


**Questions:**

- On the feedback graph model [https://arxiv.org/pdf/1106.2436.pdf], would it be correct to say that the setup in this paper can be modeled by considering at each point, the previously sampled point is removed from $G_t$ (the next feedback graph). If so, would the algorithms for sleeping expert setting be comparable? For example, the empirical evaluations in [9] are not directly comparable per my reading - if so, this can be mentioned explicitly in the text.


**Limitations:**

- Line 94. "each arm can be be pulled twice" -- per my understanding this should read "each arm {\em cannot} be pulled twice"
- Definition 1 Line 134. Request to define $H(X | Z)$ for completeness.
- Line 146. "A higher  $\lambda$ weigh" - double line.
- Line 177. Define $e_i$ (basis vector) before using it. Its' defined later in Line 187.
- Line 201. "given $x$ being selected", I understood this as $X_t = x$ (at time t, node $x$ selected for labelling which gives its value and noisy values for all its labels). Can this be clarified in writing?


**Strengths And Weaknesses:**

### Strengths
+ Good survey of related work and putting their results in context.
+ Application of IDS to linear model, low-rank matrix model and graph model.
+ Interesting real-world application (Reaction Condition Discovery)


### Weaknesses
The writing could be made a bit clearer.

- I am a bit fuzzy on the real-world application which could be better explained in the supplementary material if not the main text. Per my understanding, the dataset has been taken - PDNC or CNCCI (https://doyle.chem.ucla.edu/wp-content/uploads/2020/07/43-Predicting-Reaction-Performance-in-C-N-Cross-Coupling-Using-Machine-Learning.pdf), but the final result is not comparable to cited papers. For example, Shim etal. (https://pubs.rsc.org/en/content/articlelanding/2022/sc/d1sc06932b original ref [34]) - I would have expected to see something like Figure 3 (if comparing the IDS in the Active Learning setup). {\em Else it should be clarified that a contrived subtask has been taken to evaluate the problem and show a possible future application.} I understand this is not the main focus of the paper so that clarification would be sufficient.

- For clarity and completeness, please add a section in supplementary material explicitly describing the PDNC and CNCCI datasets (tied to the data available in this upload, namely, Informer.xlsx and doyle_matrix.xlsx),  and, how this translates to the problem setting.

- The parameter $\lambda$ controlling the ratio between instant regret and information gain is also used in the approximate algorithm (supplementary material H). However, its value is not clarified and the sensitivity of empirical results depending on its choice is unclear to me (or I missed it)?

---

> ### Author Response · Authors · 2022-07-28
> **Responses**
>
> > I am a bit fuzzy on the real-world application which could be better explained in the supplementary material if not the main text. Per my understanding, the dataset has been taken - PDNC or CNCCI (https://doyle.chem.ucla.edu/wp-content/uploads/2020/07/43-Predicting-Reaction-Performance-in-C-N-Cross-Coupling-Using-Machine-Learning.pdf), but the final result is not comparable to cited papers. For example, Shim etal. (https://pubs.rsc.org/en/content/articlelanding/2022/sc/d1sc06932b original ref [34]) - I would have expected to see something like Figure 3 (if comparing the IDS in the Active Learning setup). {\em Else it should be clarified that a contrived subtask has been taken to evaluate the problem and show a possible future application.} I understand this is not the main focus of the paper so that clarification would be sufficient.
>
> Answer: [34] is a related paper studying the problem in a transfer learning framework. It would be an promising direction to extend our results. We will include the Figure 3 for real data application in the appendix.
>
> > For clarity and completeness, please add a section in supplementary material explicitly describing the PDNC and CNCCI datasets (tied to the data available in this upload, namely, Informer.xlsx and doyle_matrix.xlsx), and, how this translates to the problem setting.
>
> Answer: thanks for your suggestions. We will add a `README.md` in the code to describe the meaning and usage of each file and add a new section in the Supplementary to describe that as well.
>
> > The parameter $\lambda$ controlling the ratio between instant regret and information gain is also used in the approximate algorithm (supplementary material H). However, its value is not clarified and the sensitivity of empirical results depending on its choice is unclear to me (or I missed it)?
>
> Answer: we use $\lambda = 2$ for linear and generalized linear models and $\lambda = 3$ for low-rank matrix and graph models.
>
> > On the feedback graph model [https://arxiv.org/pdf/1106.2436.pdf], would it be correct to say that the setup in this paper can be modeled by considering at each point, the previously sampled point is removed from $G_t$ (the next feedback graph). If so, would the algorithms for sleeping expert setting be comparable? For example, the empirical evaluations in [9] are not directly comparable per my reading - if so, this can be mentioned explicitly in the text.
>
> Answer: sleeping expert can be comparable. We will mention it explicitly in the text.

---

### Official Review · Reviewer_A6az · 2022-07-09

**Rating:** 6
**Confidence:** 3
**Soundness:** 4 excellent
**Presentation:** 4 excellent
**Contribution:** 3 good

**Summary:**

This paper introduces a sequential decision-making problem where the goal is to select a few samples and the reward is the sum of the responses from the samples. We want to maximize the reward (or minimize the regret), and we want to choose the samples adaptively, in an exploration-exploitation fashion. This is achieved by using information gain to determine the next element to sample. A number of heuristics/variants are proposed for when the information gain cannot be computed easily. An experimental evaluation, and a realistic case study, complete the paper, showing the performance of the proposed method.

**Questions:**

* In the pseudocode for the algorithm (Algorithm 1), the computation of $\Delta_t(x)$ and $g_t(x)$ is vaguely described as being done "using posterior $\phi(\cdot \mid \mathcal{F}_{t-1})$. It is very unclear to me how that would happen, even if we had complete knowledge of the posterior and could easily draw samples or compute any probability according to the posterior. I guess my issue or maybe my question is whether, in the definitions of $\Delta_t(x)$ and $g_t(x)$, the expectations are taken also w.r.t. the posterior (and if so, why is that correct). I think that, if that is the case (or whatever is the case), this aspect should be better clarified (perhaps right now is mentioned on lines 83-84, but perhaps it should be repeated closer to the definitions above.

* On line 94, I think that "be be" should be changed to "not be".

**Limitations:**

I do miss a discussion of the limitations of the proposed approach and/or of the proposed problem. It is hard to predict negative societal impact, but one could assume that such an  algorithm could be used to target individuals on the bases of some reward that could be obtained by having them perform some action, which one has to wonder whether would negatively affect underserved populations and minorities.

**Strengths And Weaknesses:**

* The presentation is quite clear. The discussion of the relation with MAB and related problem is commendable in terms of clarity. The one issue in clarity is a single but crucial aspect about the use of the posterior. See the question below.
* The proposed problem is quite interesting, and relevant to areas outside "core" machine learning.
* The proposed algorithm, with its use of information gain to minimize Bayesian regret, makes sense and it is described well. The "instatiations" of the algorithm under different knowledge of the model are also interesting and make the paper a bit more concrete.
* The evaluation is pretty convincing.
* Additional discussion of the limitations of the proposed approach would have well completed the work.

---

> ### Author Response · Authors · 2022-07-28
> **Responses**
>
> > In the pseudocode for the algorithm (Algorithm 1), the computation of $\Delta_t$ and $g_t$ is vaguely described as being done "using posterior. It is very unclear to me how that would happen, even if we had complete knowledge of the posterior and could easily draw samples or compute any probability according to the posterior. I guess my issue or maybe my question is whether, in the definitions of
> $\Delta_t$ and $g_t$, the expectations are taken also w.r.t. the posterior (and if so, why is that correct). I think that, if that is the case (or whatever is the case), this aspect should be better clarified (perhaps right now is mentioned on lines 83-84, but perhaps it should be repeated closer to the definitions above.
>
> Answer: Yes, the instant regret and information gain are computed based on posterior distributions. We will make it more clear when the abstract algorithms are introduced. In practice, calculating them directly from posterior distribution can be hard. We discuss this issue in Section 5.1.

---

> > ### Comment · Reviewer_A6az · 2022-08-05
> > **Thank you.**
> >
> > Okay, thank you for clarifying.

---

### Official Review · Reviewer_LiPB · 2022-07-11

**Rating:** 4
**Confidence:** 4
**Soundness:** 3 good
**Presentation:** 3 good
**Contribution:** 2 fair

**Summary:**

The paper tackles an active learning/bandit problem in which utility is defined to be the sum of the observed labels.
The proposed policy minimizes the ratio between the expected instant regret and the expected gain in information about the labels of the top unlabeled points (the number of these top unlabeled points matches the remaining budget).
In some settings where exact computation is intractable, the policy falls back to MCMC posterior sampling.
The authors then present theoretical results about the regret bound for their policy under specific settings such as linear models, low-rank matrix, and graph problems, and discuss situations in which we can obtain a low regret.
The paper also includes extensive experiment results showing that their method is competitive against popular baselines like UCB and Thompson sampling.

**Questions:**

It would be great if the authors could elaborate on their contributions with respect to the AS literature.
I would be interested to see a comparison between IDS and the policies in [2] and [3], which are specifically designed for AS and should serve as stronger baselines than UCB and Thompson sampling.
In particular, the policy in [3] maximizes the expected value of the sum of labels of the top unlabeled points, which should have the same computational cost as IDS but is more utility-centric.
Along the same lines, there could be variant of IDS where the numerator of the acquisition function is the _cumulative regret_, as opposed to the instant regret.
I think this would have the same cost as IDS, since you would have to pick out the top unlabeled points anyway.
I would be interested to see how this variant performs.

I would also like to see a couple of things added to the presentation of the experiment results.
- What is the interpretation for the sharp turns in the first row of Figure 1 (in the middle of Figure 1c, for example)?
- UCB is missing from some of the plots.
- The authors could consider adding information about whether results in Table 1 are statistically significant.

Miscellaneous:
- The authors could consider making the plot axis labels larger and use a larger, shared legend.
- Line 86: missing dot before "For categorical"
- Line 94: I believe the authors meant "each arm can only be pulled once."
- Line 146: Redundant "A higher lambda"

**Limitations:**

Yes, the paper mentions its limitations in the Discussion section in the form of future directions.

**Strengths And Weaknesses:**

The paper is well written and clear.
The structure of the exposition is good, and both the problems and proposed policy are well-motivated.
The authors did a good job connecting this problem to multi-armed bandit, and I enjoyed reading about the special settings like low-rank matrix and graph problems.
It is clear from their experiment results that their policy consistently performs well across many tasks compared to other benchmarks in the bandit literature.

In terms of weaknesses, I believe the authors have missed a particularly relevant line of research on the "active search" problem [1, 2, 3].
There seem to be significant similarities between ASD and AS: utility being the sum of observed labels, large amount of unlabeled data, limited labeling budget, etc.
As of now, I am not sure whether this setting is any different from those studied in the references above.
Further, although the information-theoretic approach is a common heuristic, it is not clear to me whether it actually brings additional value, compared to a variant of the policy that aims to minimize the expected _cumulative_ regret (which should roughly have the same cost as the proposed policy).
More details are included in the Questions section.

Overall, the submission addresses an important problem and the proposed solution seems to work well on the inspected settings.
The theoretical results are also interesting.
However, this work would benefit from a closer look at the AS problem and highlighting why their approach of optimizing their acquisition function is novel and beneficial.
This would help situate the paper in the literature better and clarify its contributions.

[1] Garnett et al. Bayesian Optimal Active Search and Surveying. ICML 2012.

[2] Vanchinathan et al. Discovering valuable items from massive data. KDD 2015.

[3] Jiang et al. Efficient Nonmyopic Active Search. ICML 2017.

---

> ### Author Response · Authors · 2022-07-28
> **Connections to Active Search Literature**
>
> We thank the reviewer for pointing out the important literature on Active Search (AS). Active Search considers the same formulation as ASD. However, we argue that our paper makes significant contributions to the literature.
>
> **Comparing IDS and ENS**
>
> We first discuss the similarity between IDS and an important method in AS literature, ENS (efficient nonmyopic search) and argue that it is significant to introduce IDS to the literature. As we discussed in the paper, IDS tries to balance between exploitation and exploration by minimizing the ratio of instant regret and information gain on the top $T-t$ points at the step $t$. There is a variant of IDS that considers a weighted sum of the two, i.e. $\Delta^T \pi + \gamma g^T \pi$ for some constant $\gamma$.
>
> ENS targets at a similar exploration and exploitation as discussed in [3]. Using the notations in [3], ENS maximizes
>
> $Pr(y_t = 1 \mid x_t, \mathcal{D}_{t - 1}) + E\_{y_t}[\sum\_{T-t}^{'} Pr(y = 1 \mid x, \mathcal{D}\_{t})]$,
>
> Since $E[\sum\_{T-t}^{'} Pr(y = 1 \mid x, \mathcal{D}\_{t-1})]$ is constant for different choices of $x_t$, maximizing the above term is  equivalent to minimize
>
> $1 - Pr(y_t = 1 \mid x_t, \mathcal{D}_{t - 1}) (term (1)) + E[\sum\_{T-t}^{'} Pr(y = 1 \mid x, \mathcal{D}\_{t-1})] - E\_{y_t}[\sum\_{T-t}^{'} Pr(y = 1 \mid x, \mathcal{D}\_{t})] (term (2))$.
>
> The term (1) $1 - Pr(y_t = 1 \mid x_t, \mathcal{D}_{t - 1})$ is the instant regret as in IDS. The term (2) represents the extra gain brought by labeling $x_t$ at the step $t$. One can see that term (2) depends on the change of posterior distribution of the top $T-t$ points altering observing $y_t$, which is a reflect of the KL-divergence of $\phi(\cdot \mid \mathcal{D}_t)$ and $\phi(\cdot \mid \mathcal{D}\_{t-1})$. Lemma 2 shows that information gain can be used to upper bound term (2). Thus, we two methods are intrinsically connected.
>
> Now we discuss the main difference between IDS and ENS. We observed that ENS tends to over explore in the experiments on linear models. See our new Figure 9 in the appendix. We believe this is because ENS assumes that the labels of all remaining unlabeled points are conditionally independent. That is the extra gain by observing the new label $y_t$ is uniform across all the remaining $T-t$ points. This is over-estimating the gain, because in the later stage when the estimates on $Pr(y = 1 \mid x)$ are more accurate, the extra gain from observing a single label is also much less. ENS, thus, weighs too much on the exploration side. We highly believe that this will lead to linear regret instead of $\sqrt{T}$ regret that can be achieved by IDS. This is also reflected in Figure 9, where ENS performs worse than IDS when the noise level of the problem is low and we need more exploitation. In general, IDS provides a more flexible balance between exploration and exploitation.
>
> **Summary of contributions to Active Search**
>
> We give a brief summary of contributions of our work to the Active Search literature.
> - We introduce IDS that is substantially different from previous methods in the literature. We show a generic regret bound for IDS and discussed the actual forms of the regret bound for three interesting cases that have not been theoretically studied in the literature. To my best of knowledge, the only work that applies bandit algorithm with a regret guarantee is [2], which considers a different Gaussian Process model.
> - We first apply the algorithm to reaction condition discovery model, which achieves significant improvement over baseline methods.
>
> [3] Jiang et al. Efficient Nonmyopic Active Search. ICML 2017.
> [2] Vanchinathan et al. Discovering valuable items from massive data. KDD 2015.

---

> > ### Author Response · Authors · 2022-07-28
> > **Responses to other questions**
> >
> > > Along the same lines, there could be variant of IDS where the numerator of the acquisition function is the cumulative regret, as opposed to the instant regret. I think this would have the same cost as IDS, since you would have to pick out the top unlabeled points anyway. I would be interested to see how this variant performs.
> >
> > Answer: we can implement that variant, but we don't understand the intuition of using cumulative regret. The future cumulative effects are given in the information gain term as we discussed in the comparison between IDS and ENS. The sum of all the $T-t$ future gain will over-estimate the effects of exploration as we discussed above.
> >
> > > What is the interpretation for the sharp turns in the first row of Figure 1 (in the middle of Figure 1c, for example)?
> >
> > Answer: In Figure 1a, the phase change happens when the number of samples reaches the number of parameters, before which the estimates can be arbitrary. In Figure 1c, the calculate the regret with respect to the actual labels that are either $1$ or $0$. As a result, the first half of regrets are compared to all 1's and the second half is compared to all 0's.
> >
> > > UCB is missing from some plots: we only tested UCB for linear and generalized linear models.
> >
> > Answer: It is not clear how to implement UCB for low-rank matrix model. There is no trivial solution to that. That is why we only compared IDS with TS and random.
> >
> > > The authors could consider adding information about whether results in Table 1 are statistically significant.
> >
> > Answer: thanks for the suggestions. We will add that information to the table.
> >
> > All the other typos are fixed in the revised version.

---

> > ### Comment · Reviewer_LiPB · 2022-08-04
> > **Response to the authors**
> >
> > Thank you for the responses, especially the detailed discussion of the differences between IDS and ENS.

---

### Meta-Review · Area_Chair_fBkY · 2022-08-30

**Recommendation:** Accept
**Confidence:** Less certain

**Metareview:**

This was a boredline paper.
However, the reviewers like it, and it seems that the authors answered all the concerns of Reviewer LiPB and myself.
Please add the comparison of IDS and ENS to the final version.


**Award:**

No

---

### Decision · Program_Chairs · 2022-09-14

Accept